# A Cost Analysis of Diabetic Hand Infections: A Study Based on Direct, Indirect, and One-Year Follow-Up Costs

**DOI:** 10.3390/healthcare13151826

**Published:** 2025-07-26

**Authors:** Burak Kuşcu, Kaan Gürbüz

**Affiliations:** 1Special Bandırma Royal Hospital, Balıkesir 10200, Türkiye; 2Kayseri Kızılay Hospital, Kayseri 38000, Türkiye; drkagangurbuz@gmail.com

**Keywords:** diabetes mellitus, complications, economic, burden, healthcare

## Abstract

**Background**: Diabetes mellitus is a chronic metabolic disorder that increases mortality and morbidity rates. Infections of the hand can easily cause long-term morbidity and dysfunction, but despite their associated high morbidity, diabetic hand infections are more neglected than diabetic foot infections. **Objectives**: This study was conducted over a one-year follow-up period, considering the total costs of treatment over one year post discharge for patients with diabetic hand infections that required surgery. A Monte Carlo Simulation was used in this study as a sensitivity analysis of all the cost calculations. **Materials and Methods**: A total of 62 out of 75 patients were diagnosed with Type 2 diabetes; 11 were female, and 64 were male. Out of all the patients, 15 visited outpatient clinics 30 times or more, and due to their recurrent visits, the outpatient treatment costs reached USD 5162.41 ± 3838.55. The total cost incurred over the period from the patients’ first hospitalization to the completion of all treatments and the end of the one-year follow-up was USD 24,602.22 ± 5257.15. **Conclusions**: The cost of hospitalization was the most important factor affecting the total expenses. Therefore, taking precautions before a diabetic hand infection occurs, or when one does occur, performing treatment without delay is expected to reduce the economic burden.

## 1. Introduction

Diabetes mellitus is a chronic metabolic disorder that increases mortality and morbidity rates. Its incidence is likely to double between 2005 and 2030, according to the World Health Organization, which estimates that 346 million people worldwide have diabetes [1]. According to the International Diabetes Federation Diabetes Atlas (2025 Edition), approximately 537 million adults aged 20−79 years are living with diabetes globally, and this number is projected to rise to 783 million by 2045 [2].

Diabetic hand lesions, while less frequently studied than diabetic foot ulcers, pose a significant health burden. Their incidence is estimated to be approximately 0.37−1.0% among patients with diabetes. These infections are commonly associated with poor glycemic control, peripheral neuropathy, and immunosuppression. The clinical outcomes can range from superficial cellulitis to deep-space infections requiring surgical intervention. Recent studies have highlighted the role of delayed presentation and a lack of early surgical consultation as key contributors to morbidity and prolonged hospitalization [3].

In patients with diabetes, infection could be simply described as the result of the balance between host resistance and microorganisms being disrupted in favor of the latter, microorganisms’ evasion of the immune system, and their proliferation in the system [4]. Even a simple skin infection can go as far as osteomyelitis, as the ability to heal decreases. 

Infections of the hand can easily cause long-term morbidity and dysfunction [5]. Despite their associated high morbidity, however, diabetic hand infections are more neglected than diabetic foot infections [6]. Consequently, the treatment of diabetic hand infections is often delayed, owing to a lack of knowledge of the possible risk or a reliance on conventional treatments [7]. Although joint mobility limitations, Dupuytren’s contraction, and trigger finger are the most common ailments in the hands of patients with diabetes, ulcers due to diabetic infections can easily cause serious complications that can lead to amputations [8].

Diabetes is one of the most costly and burdensome chronic illnesses ever recorded [9]. In 2014−15, the economic burden of diabetes in England was approximately USD 1.4-USD 1.6 billion, and it is expected to be USD 14.7 billion higher in 2030 [10]. Hand infections are an important cause of health expenditures. Even if diabetic hand infections are treated appropriately, both the expenditures associated with the treatment process and scar tissue, joint contractures, stiffness, chronic pain, amputation, and the subsequent loss of labor after treatment can contribute to the economic burden.

Several studies in the literature record the incidences and outcomes of diabetic and/or non-diabetic hand infections [11,12]. However, none of them describe the economic burden of diabetic hand infections. Thus, our study aimed to raise awareness of this economic burden and draw attention to the importance of preventive medicine by showing the economic consequences of diabetic hand infections and their complications, which are absent in the literature or less emphasized than diabetic foot infections.

## 2. Material and Methods

After obtaining ethical approval from local ethics commissions, a prospective, open-label, non-randomized observational study (i.e., case series) was conducted over one year. Data (sourced from the Hospital Information Management System) was collected descriptively and cross-sectionally on all the diabetic hand infections surgically treated in our diabetic foot and chronic wound care clinic from November 2015 to November 2020. Prior to their enrollment, all the patients provided informed written permission, and the research was performed in compliance with the Helsinki Declaration’s principles.

Patients were included if they met all of the following criteria: (1) infections located in the fingers, thumb, hand, wrist, or forearm that required surgical intervention; (2) wound cultures obtained at the time of diagnosis; (3) the initiation of antibiotic therapy prior to surgery; and (4) a minimum follow-up period of one year after the diagnosis of diabetes mellitus and/or a diabetic hand infection, with complete medical records kept. The exclusion criteria included infections located above the elbow joint and postoperative infections. In addition, patients with incomplete medical records or those lost to follow-up were excluded from the analysis.

Preoperatively, we recorded the patients’ name, age, sex, file number, dominant hand, type of diabetes, glycated hemoglobin (HbA1c), white blood cell (WBC) count, sedimentation level, and C-reactive protein (CRP) level, along with the side operated on, the location and anamnesis of the hand infection, the type of diabetes, co-existing morbidities, and the date of surgery. Perioperatively, we recorded the type of surgical intervention (ray amputation, open amputation, drainage, vacuum-assisted closure (VAC), fasciotomy), complications and their treatments, the gram culture, the length of the hospital stay, and follow-up data for at least one year. Also, the total cost of treatment and the costs immediately and at a follow-up one year post discharge were calculated. All the costs were converted to their equivalent in the last quarter of the year 2022 according to the European Central Bank foreign exchange reference rates. Furthermore, a Monte Carlo Simulation was performed to conduct a sensitivity analysis in this research. 

A Monte Carlo Simulation (MCS) was conducted to estimate the long-term economic burden associated with diabetic hand infections and to model the impact of variable clinical and economic parameters on the overall cost. This simulation was not used to analyze retrospective clinical data but was implemented as a complementary tool to project future cost scenarios under uncertainty. The primary aim of the MCS was to explore how the variations in key input parameters—such as the hospitalization rates, recurrence of infection, antibiotic therapy duration, surgical intervention rates, and unit healthcare costs—affected the total direct and indirect costs over a one-year follow-up period. The key input variables were identified from both our dataset and the relevant literature. Where available, empirical distributions were derived from the observed data. Otherwise, expert opinions and published estimates were used [13]. The following probability distributions were applied:Hospitalization cost: A triangular distribution (min: TRY 8000; mode: TRY 12,000; max: TRY 18,000).Surgical intervention cost: A normal distribution (mean: TRY 6500; SD: TRY 1200).Antibiotic therapy duration (days): A uniform distribution (range: 10–21 days).Re-infection rate: A beta distribution (α = 3, β = 20), based on recurrence observations.Lost workdays: A triangular distribution (min: 7; mode: 14; max: 28 days).

A total of 10,000 iterations were run for each scenario by randomly sampling from the assigned distributions. The results were summarized as the mean total costs with 95% confidence intervals. A probabilistic sensitivity analysis was also performed to assess the robustness of the findings across a range of input uncertainties. We chose to perform an MCS due to its ability to incorporate parameter uncertainty and provide a probabilistic range of expected outcomes. Compared to deterministic models, this approach provides more realistic scenario analysis and policy-relevant insights, especially in the context of healthcare cost forecasting, where the input variability is high, and was designed according to established principles in probabilistic modeling [13].

The primary outcomes of this study were defined as (1) the direct treatment costs; (2) indirect productivity losses due to an incapacity for work; and (3) cumulative one-year follow-up costs. 

The cost analysis included multiple categories of medical and non-medical expenses. These were costs associated with hospitalization (ward and ICU stays), medications (antibiotics, insulin, and analgesics), surgical procedures (debridement and amputation), outpatient clinic visits, imaging and laboratory tests, and rehabilitation sessions, as well as indirect costs associated with a loss of work productivity due to treatment or disability.

The cost analysis in this study was conducted from a healthcare payer perspective using a bottom-up costing approach, while also incorporating the indirect costs of lost productivity through a human capital approach. Although a societal perspective is implied by the inclusion of productivity losses, it was not comprehensively adopted by this study. The indirect costs were estimated by applying the national average wage loss to the number of missed workdays [14]. 

All the cost data were converted to US dollars using the European Central Bank’s average annual exchange rates and adjusted for inflation using the US Consumer Price Index (CPI) for the last quarter of 2022. This method was consistently applied across all the direct and indirect cost categories.

The cases in this research were analyzed over a span of ten to fifteen years, and the expenses were presented in terms of the cost/patient/year in Euros and British Pounds. In addition, invoices from the Turkish Social Security Service with amounts in Turkish Lira were used to perform the present research. To enable comparisons, these expenses needed to be converted into a consistent currency. As an accessible methodology, the authors decide to convert all the expenses analyzed and reported in this study into US dollars. For this purpose, the steps below were followed:Using the European Central Bank’s (htpp://www.ecb.europa.eu, accessed on 23 July 2025). annual average foreign exchange rates, a multiplicative factor for converting the cost to the corresponding year’s US dollar value was calculated.The cost was then multiplied by the conversion factor to obtain a USD amount.According to the US Annual Inflation statistics, this amount was converted to its equivalent in the final quarter of 2022 (https://www.bls.gov/data/inflation_calculator.htm, accessed on 23 July 2025).

In this study, a sensitivity analysis was carried out through the use of a Monte Carlo Simulation (MCS). MCS is an approach that uses formulas in favor of random number generation and iterative trial and error. Due to its statistical modeling capabilities, an MCS is able to offer a rough estimate of the range of probable expenditures. Concurrently, the cost distribution variables are obtained. This simulation approach is not a theory but rather a tool for finding answers to difficult questions. Problems can be tackled in a variety of ways using this method, all of which are dependent on the specifics of the system and the model that will be built from it. An observational distribution of the variance is used to produce a sample during the simulation procedure, and random numbers are used to represent improbable circumstances. An MCS is capable of giving all the parameters and variables random values in line with the probabilities. It is this random number generation that forms the basis of the simulation. For instance, the probability distribution, based on values randomly picked from a normal distribution, looks like this for a 66% chance that treatment will be needed for osteomyelitis: if the patient’s random number is between 00 and 66, they will receive osteomyelitis therapy, and if it is between 67 and 99, they will not require any special care for their condition. In this research, the distribution parameters were calculated at the diabetic hand infection diagnosis, treatment, and follow-up stages. As part of this study, we simulated every conceivable scenario five thousand times and used the data to produce distribution values with a one-year horizon.

## 3. Results

The research comprised 75 participants followed up for at least one year and complete data from 173 individuals who matched the inclusion criteria. A total of 11 were female with a mean age of 61.5 ± 8.12, 64 were male with a mean age of 56.2 ± 11.3, and 62 patients were diagnosed with Type 2 diabetes. Similarly, in the 75 patients included in the study, the infection-causing microorganism was isolated, and the majority of cases were caused by a mixed-type infection, as shown in detail in Figure 1.

Of all the patients, 59 were right-handed, and 48 of the right-handed patients and 6 of the left-handed patients had an infection in their right hand. While the dominant hand was infected in 58 of the patients, the non-dominant hand was infected in 17. The distribution of the infection locations is shown in Figure 2.

On a clinical examination at admission, the patients presented with various signs of infection, including purulent discharge (52%), erythema (68%), localized or diffuse swelling (74%), gangrene (29%), and limited joint motion (35%). These findings helped to characterize the severity and extent of the diabetic hand infections evaluated in this study.

The primary metabolic blood test results at the time of diabetic hand infection diagnosis are shown in Table 1, and the diagnoses in all the cases according to their ICD-10 (International Classification of Diseases, Tenth Revision) codes are shown in Figure 3.

The total cost for the treatment of the patients in this study included the costs incurred during their 1-year follow-up period and whole hospitalization duration. The components that constituted this cost were the outpatient clinic visit costs, public health insurance report fees, test/examination and radiology costs, and antidiabetic drug costs. In Table 2, detailed costs for every step of the treatment process are given.

A total of 15 patients visited outpatient clinics 30 times or more, and the outpatient treatment costs increased due to the recurrent visits of these patients. The total cost incurred from the patients’ first hospitalization to the completion of all treatments and the end of their outpatient clinic visits is also shown in Table 1. On one hand, the relationship between the number of outpatient clinic visits and the surgical procedure type was not statistically significant (*p* = 0.151), as shown in Table 3. On the contrary, a statistically significant relationship was found between the type of surgical intervention and the outpatient clinic visit costs (*p* = 0.004), as shown in Table 4.

## 4. Discussion

This study analyzed the direct and indirect healthcare costs associated with diabetic hand infections over a one-year period. The primary findings indicate that hospitalization and delayed treatment were the main contributors to the total cost, while the type of surgical intervention and the clinical severity of the infection influenced the cost variability. These outcomes directly fulfill the research objective outlined in the Introduction: to quantify the economic burden of diabetic hand infections and identify potentially modifiable cost drivers that can inform early intervention strategies.

Although it is known that foot ulcers are often related to diabetes due to neuropathy and ischemia, the number of hand ulcers is likely to increase as the number of diabetic patients is expected to double over the next decade, reaching approximately half a billion. It is fairly unusual for a superficial infection to lead to amputation in this patient population, but the risk is still at least eleven times higher [15,16,17]. Therefore, the process of treating diabetic hand infections, unlike other complications of diabetes, should begin at the moment of diagnosis and even be performed prophylactically. 

Diabetic hand lesions are not formally classified as chronic complications of diabetes by most professional organizations (e.g., the ADA and EASD) but are instead considered acute conditions that require urgent medical intervention due to their potential for rapid progression and severe outcomes [18].

In the region where this study was conducted, admission to the nearest health institution for blood glucose level regulation in patients with diabetes is unfortunately very rare [17]. In fact, eleven patients in our study were unaware that they had diabetes before the infection developed in their hands. In the literature, late presentation to a health institution is reported as a poor prognostic factor in upper extremity infections related to diabetes [19].

The economic burden of diabetes and its complications, especially the impact of foot ulcers on healthcare systems and the individuals affected, is significant [20,21,22,23,24]. However, the current literature does not provide any information about the economic burden of diabetic hand infections. Our study is the first to reveal this over a five-year period.

The situation has been highlighted by recent studies regarding the economic burden on healthcare systems and individuals with diabetes caused by foot-related complications, including the cost of both outpatient and inpatient treatment. Some examples of studies from all over the world include the following. Firstly, in England, the National Health Service (NHS) cost for the year 2014–2015 was USD 1.4–1.6 billion. This represented over 1% of the health budget, or USD 1.25 of every USD 175 spent by the NHS [10]. Despite the fact that about 90% of this expense was due to foot ulcers, there is no available data for diabetic hand infections, ulcers, and/or amputations. In England, the total cost of care, including outpatient and inpatient treatments, for a cured foot ulcer was estimated to be USD 2645; for an unhealed ulcer to be followed up, it was USD 10,865, and for an amputation due to an ulcer, it was USD 20,955 [25]. Infections of foot ulcers during outpatient treatment markedly raised the costs in England to USD 16.075 [26].

According to a study conducted in Russia, the average expenditures to treat patients admitted to a hospital with foot ulcers based on their Wagner Classification were as follows: USD 2600 for grade 1, USD 2995 for grade 2, USD 4179 for grade 3, and USD 5668 for grade 4. Similarly to our results, the length of the hospital stay, foot surgery, and vascular surgery had the highest impacts on the cost [27]. Similar results demonstrating that the average cost increased as the Wagner Classification grade of an ulcer worsened were also shown by Acar et al. [28].

This economic burden is even worse in the USA. Between 2006 and 2010, 1,019,861 patients with diabetic foot ulcers were submitted to emergency departments, representing 1.9% of the total 54.2 million cases of diabetes. The national cost for the emergency departments was USD 1.9 billion to 8.78 billion per year (USD 2014) for 81.2% of the admitted patients, including inpatient expenses. The clinical outcomes included death in 2.0% of cases, sepsis in 9.6% of cases, and amputation in 10.5% of cases (a major–minor amputation ratio of 0.46) [29]. The cost of diabetic foot management in the USA between 2007 and 2011 was approximately USD 9–13 billion. An additional USD 11,710–16,833 might have been added to a patient’s yearly healthcare expenses due to diabetic foot ulceration, doubling the cost of providing diabetes treatment [30]. There were also large indirect costs, such as the loss of individual wages, the impact on patients’ careers, and the repercussions of absenteeism for employers. In 2017, diabetes treatment directly cost the United States USD 237 billion, with one-third of these expenses attributable to diabetic foot ulcers [31].

Lastly, in a systemic review conducted by Tchero et al. comparing the diabetic foot infection costs in five European regions, it was reported that although all the expenses were substantial and varied considerably across Europe due to disparities in the provision of and access to healthcare, it was certain that the economic burden of diabetic foot infection had increased in the last few decades and will increase much more over the next two decades [32]. The literature published up to now shows the presence of diabetic hand infection cases, not just diabetic foot infection cases, in the invisible part of the iceberg and the associated high health expenditures that awaits healthcare systems.

An observational study conducted in Turkey was published in 2016 and was similar to ours regarding the average total cost of a diabetic foot ulcer per patient, USD 14,146.8. Also, in 2014, the health expenses per person in Turkey were USD 1045, and the average annual per-patient cost to treat diabetic foot ulcers was 14 times that amount [33]. Ten years later, we found that the economic burden, not of diabetic foot ulcers but of diabetic hand infections, at this time and their complications reached USD 24,602, 24 times the health expenses per person in 2014.

The results of this study are congruent with those of previous ones on the cost analysis of diabetic foot ulcers and their complications. In Turkey, the largest proportion of expenditures related to diabetic hand infections and their complications have arisen from inpatient clinic treatments. This should encourage the implementation of preventive approaches. The distinguishing characteristic of the current study is that the expenditures per patient were approximately the same for diabetic hand infections and their complications as for diabetic foot ulcers. 

Regarding the limitations of our study and future directions, the budget for diabetic foot and/or hand infection treatment and care is set by each country’s government and strictly depends on the country’s level of economic development. There are also certainly disparities in the methods that can be used for calculating the costs. The costs estimated here do not represent those for the treatment of all diabetes patients in Turkey, and this study was limited to analyzing the expenditures incurred within a one-year follow-up period and conducted in just one hospital. There is thus an urgent need for reliable and practicable national guidelines including uniformly calculated components of all the costs, which will aid in assessing the economic burden of diabetic hand infections and their complications. Further, this will aid in the identification of high-value resources whose consumption may be reduced using cost-effective healthcare strategies. 

Although a post hoc power analysis was initially considered, the retrospective design and sample size limitations of this study did not allow for a robust statistical power calculation. Therefore, the possibility of Type II errors in the subgroup comparisons remains a relevant limitation.

Additionally, data on confounding variables, such as patients’ socioeconomic status and delays in treatment, were not systematically available for all the patients. Therefore, an adjustment for these variables could not be conducted in the analysis. This limitation should be addressed in future multicenter studies through structured data collection and multivariable statistical modeling. Additionally, although a statistically significant relationship was observed between the surgical intervention types and outpatient costs (*p* = 0.004), this study was not powered or designed to permit a formal cost-effectiveness comparison. Future prospective studies should be conducted to evaluate the comparative economic value of different surgical strategies. Finally, cost comparisons between the treatment of diabetic hand infections and diabetic foot ulcers across countries must be interpreted with caution. Due to variations in healthcare systems, cost-reporting standards, and economic conditions, these comparisons are meant to be illustrative rather than inferential. This limitation has been noted to avoid overgeneralization in cross-national cost interpretation.

## 5. Conclusions

The main factor that increased the cost was the treatment applied during the period prior to the need for surgery and the duration of its implementation. To mitigate future increases in the economic burden of diabetic hand infections and their complications, the most realistic and conclusively evidenced approach will be to consult a surgeon where surgical necessity is suspected and for surgeons to perform the most appropriate and radical surgical intervention without delay. This study highlights the role of hospitalization as the main cost driver in diabetic hand infections, with treatment delays contributing to significantly higher costs. While this does not constitute a formal cost–benefit analysis, the findings suggest that early surgical referral and intervention may reduce the economic burden. We emphasize this as a policy-relevant recommendation and encourage further studies to investigate the health–economic impacts of preventive and timely care strategies in this patient population.

## Figures and Tables

**Figure 1 healthcare-13-01826-f001:**
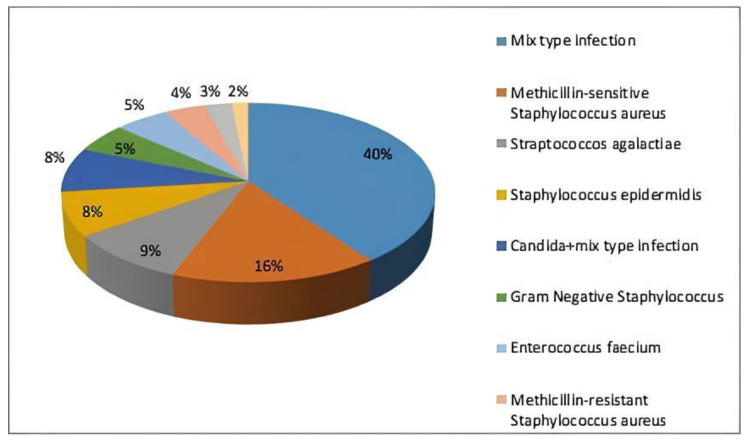
Distribution of isolated microorganisms.

**Figure 2 healthcare-13-01826-f002:**
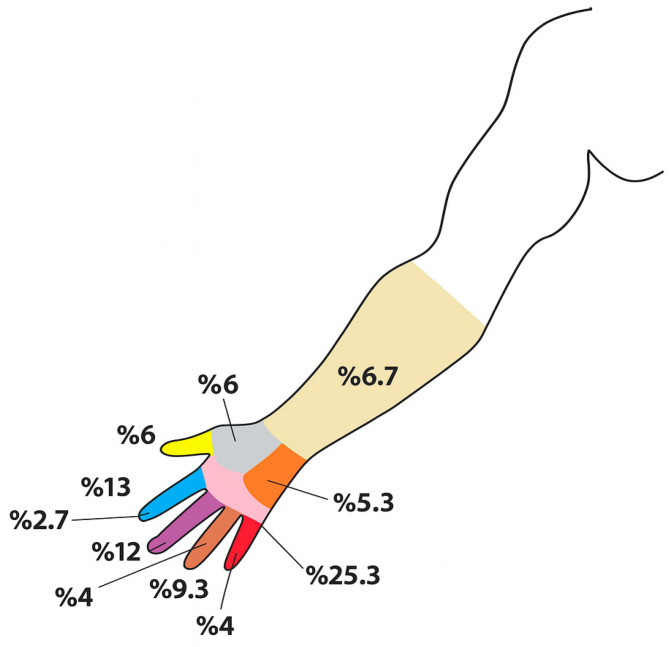
Distribution of diabetic hand infection locations at time of diagnosis.

**Figure 3 healthcare-13-01826-f003:**
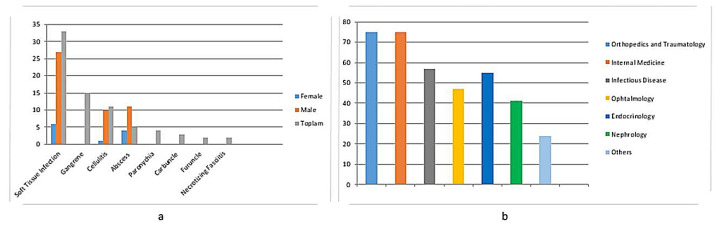
(**a**) Diagnoses in all the cases according to the ICD-10 codes and sex. (**b**) Number of outpatient clinic visits according to the clinic.

**Table 1 healthcare-13-01826-t001:** Parameters regarding the primary metabolic blood test results at the time of diabetic hand infection diagnosis.

Parameters	$\bar{x}$ ± SD/Average (Min–Max)
Age	57.01 ± 11.09/62 (28–68)
Pre-infection HbA_1_C (mmol/mol)	8.20 ± 1.33/8.10 (6−12)
Follow-up period (months)	19.13 ± 10.48/17 (1−44)
Blood sugar level at the diagnosis (mg/dL)	335.81 ± 88.14/324.0 (216.0−365.0)
At diagnosis HgA_1_C value (mmol/mol)	8.27 ± 1.15/8.30 (6−13)
Sedimentation rate at the diagnosis (mm/h)	33.75 ± 10.16/34.00 (16–65)
CRP at the diagnosis (mg/L)	49.83 ± 15.72/50.00 (25–102)
WBC at the diagnosis (#/mcL)	11,440.80 ± 3151.65/11,440.00 (2400–19,820)
Period of hospitalization (days)	22.68 ± 13.89/22.00 (0–65)

Abbreviations: SD, standard deviation; min, minimum; max, maximum. #/mcL: Number of cells per microliter.

**Table 2 healthcare-13-01826-t002:** Subtotal and total costs of diabetic hand infection treatment.

Parameters	$\bar{x}$ ± SD/Average (Min–Max)
Drug cost ^Ψ^ (US dollar)	5542.426 ± 3519.60/4473.00 (1006.00–19,873.00)
Length of stay in the clinic (days)	19.36 ± 10.45/21 (0–48)
The clinic stay cost ^γ^ (US dollar)	6222.64 ± 4377.76/5534.00 (0–18,473.00)
Length of stay in the ICU (days)	3.31 ± 6.80/0 (0–30)
The ICU stay cost ^∂^ (US dollar)	2256.09 ± 388.44/1842.5 (0–18,567.00)
of outpatient clinic visits	22.29 ± 11.01/20 (7–59)
Outpatient clinic visit cost ^ϕ^ (US dollar)	5162.41 ± 3838.55/4382.00 (384.00–15,635.00)
Subtotal without surgical treatment	6228.20 ± 4570.48/4982.00 (498.00–21,116.00)
Surgical treatment cost (US dollar)	3607.33 ± 5290.70/521.00 (21.00–19,407.00)
of Days to report incapacity for work (days)	7.85 ± 21.69/0 (0–102)
Incapacity for work cost (US dollar)	745.53 ± 203.32/0 (0–1002.00)
Total cost including whole treatment process (US dollar)	24,602.22 ± 5257.1523/21,155.00 (3166.00–68,975.00)

^Ψ^ Including all medication costs during hospitalization and the one-year follow-up period. ^γ^ Including consultations and surgical and non-surgical interventions performed by departments other than Orthopedics, such as Ophthalmology and Nephrology. ^∂^ Some, but not all, patients were admitted to the ICU not because of surgical complications but due to metabolic reasons. ^ϕ^ Including all follow-up costs, such as those for orthosis and prosthesis devices, laboratory tests, imaging, hyperbaric oxygen treatment, physical therapy and rehabilitation, and percutaneous transluminal angioplasty.

**Table 3 healthcare-13-01826-t003:** Cross-tab of number of outpatient clinic visits by type of surgical intervention.

**# of Outpatient Clinic Visits**	**Types of Surgical Interventions**	**#**	$\bar{x}$ **± SD** **Average (Min–Max)**
	Drainage	24	18.91 ± 11.13 17.00 (7.00–59.00)
	Drainage + VAC	4	18.75 ± 2.98 18.00 (16.00–23.00)
	Open Amputation + VAC	5	27.80 ± 11.94 16.50 (10.00–41.00)
	Ray Amputation	18	22.83 ± 11.28 20.50 (12.00–58.00)
	Fasciotomy + VAC	7	21.71 ± 8.57 22.00 (10.00–35.00)
	Fasciotomy + Ray Amputation	10	21.90 ± 9.42 19.50 (10.00–41.00)
	Amputation + Flap Reconstruction	6	33.50 ± 13.32 31.50 (16.00–54.00)
	Fasciotomy + VAC + Flap Reconstruction	1	21.00
	K-W	10.728, *p* = 0.151	

# indicates the number of patients undergoing each surgical intervention. Kruskal–Wallis test compares the number of outpatient visits across different surgical intervention groups. Abbreviations: SD, standard deviation; min, minimum; max, maximum; VAC, vacuum-assisted closure.

**Table 4 healthcare-13-01826-t004:** Cross-tab of cost of outpatient clinic visits by type of surgical intervention.

**Cost of Outpatient Clinic Visits (US Dollars)**	**Types of Surgical Interventions**	**#**	$\bar{x}$ **± SD** **Average (Min–Max)**
	Drainage	24	3256.45 ± 3250.38 2273.00 (384.00–12,584.00)
	Drainage + VAC	4	6489.00 ± 4541.81 7611.50 (498.00–10,235.00)
	Open Amputation + VAC	5	5434.20 ± 4095.99 4872.00 (1804.00–12,390.00)
	Ray Amputation	18	529.33 ± 229.38 457.50 (192.00–1584.00)
	Fasciotomy + VAC	7	6289.71 ± 5044.81 4409.00 (2009.00–15,635.00)
	Fasciotomy + Ray Amputation + VAC	10	4065.30 ± 2394.61 4434.50 (845.00–8943.00)
	Amputation + Flap Reconstruction	6	1010.00 ± 504.76 1291.00 (387.00–1034.00)
	Fasciotomy + Flap Reconstruction	1	1329.00 ± 204.06 1953.00 (237.00–1034.00)
	K-W	21.162, *p* = 0.004	

# indicates the number of patients undergoing each surgical intervention. Kruskal–Wallis test compares the number of outpatient visits across different surgical intervention groups. Abbreviations: SD, standard deviation; min, minimum; max, maximum; VAC, vacuum-assisted closure.

## Data Availability

For academic purposes, the chief researcher can make the data available on demand by email.

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
