# Peer review of "A Cost Analysis of Diabetic Hand Infections: A Study Based on Direct, Indirect, and One-Year Follow-Up Costs"

_healthcare, 2025, doi:10.3390/healthcare13151826_

Round 1
Reviewer 1 Report
Comments and Suggestions for Authors
The submitted journal article, titled "Cost Analysis of Diabetic Hand Infections: A Study Based on Direct, Indirect, and a One-Year Follow-Up Costs [sic]" specified its aim to 'raise awareness about the economic burdens of diabetic hand infections and to draw attention to the importance of preventative medicine showing the economic consequences of diabetic hand infections and its complications, which are none to less emphasized in the literature than diabetic foot infections [sic].' The study’s objectives in the Abstract section appear to more closely match the study’s true purpose, as the stated ‘aim’ and subsequent study methodology do not address in a scientifically meaningful way the correlate of prevention in the diabetic hand. Stated differently, the implications of this work may address prevention, but the empirical analysis itself does not. Please note that the study’s specific purpose or objective should directly lead to testable hypotheses; this study’s ‘aim’ (a term reserved for proposals) does not meet these basic criteria. Overall, while the research objective/purpose would be of interest to clinicians, payers, and the scientific literature, numerous overriding concerns limit any publication of this work in current form.
Specific points:
The study design is not described correctly (i.e., ‘a retrospective review of prospectively followed-up patients’). This appeared to be a prospective, open label, non-randomized study.
Inclusion criteria needs to be explicitly stated as such. What exclusion criteria were present? Was this an intention to treat (ITT) study? How were drop-outs managed and analyzed?
The a priori comparator groups and outcomes were not clearly articulated within the Methods section.
A key component of this research involves costing. However, the costing methods used were not described adequately, nor appropriately. For example, what was the perspective? Were activity-based costing (ABC) methods used, bottom-up, or top-down? Indirect costs did not appear to be calculated correctly, as the study did not define which specific valuation approach was utilized (nor was a sensitivity analysis conducted based upon any assumptions of that chosen valuation method). While foreign exchange reference rates were used, what inflation measure was employed? This reviewer understands that the measurement of both direct and indirect costs is one that is quite challenging, but affirms that transparency and adherence to a current research standard remains tantamount.
What comorbid conditions were present in these patients? Differences in case-mix is a critical consideration in any assessment of outcomes for a non-randomized investigation, an issue which would also require appropriate statistical control via multivariable analysis. In more detail, the investigation does not include relevant information to ensure predictive accuracy and interpretation, chiefly by the overt omission of multivariable control for demographics, case-mix risk-adjustments, confounders (e.g., acute exacerbations, smoking), or comorbidities. A specific example of the problem of confounding in this study involves the separate groups of diabetic hand infections. In the diabetic foot, for example, osteomyelitis is present in approximately 10%-15% of moderate and in 50% of severe cases. The authors do not indicate if multiple conditions were controlled for, but it is implied that it was not due to the bivariable approach utilized.
A power analysis must complement the research. It was surprising that this was not included at the onset, given that this study was prospective in nature and that the patient enrollment sample size was inherently contingent upon this. If the power of this study is low, the work may inherently be deemed unpublishable in its current context (i.e., inflated Type II errors, false positives, wide confidence intervals).
As the Monte Carlo analysis was not specified to any reasonable level directly in the Methods section, a specific reference to the study’s footnote should appear in the text iteself. Even after reading this footnote, however, it was not entirely clear how this analysis was incorporated in the study. Was the MCS used for future forecasting and not for the retrospective analysis? If this is correct, the probability distributions of treatment algorithms should be specified in more transparent and empirically-validated ways (e.g., probabilistic sensitivity analysis, distributional definitions for varying probability nodes). If not, results may be scientifically invalid and henceforth, spurious. Additionally, what justification would surround the use of MC approach, if this was employed for the retrospective analysis, vis-à-vis others? Under the conditions of simulation, it is suggested that some extended degree of variance be included surrounding the study's inputs and assumptions [either through a series of one-way sensitivity analyses coupled with a method of imputed stochastics (e.g., probabilistic sensitivity analysis, Bayesian analysis)]. Statistical analyses conducted via “descriptive statistics” are also insufficient when multivariable methods are required, and when control for multiple comparisons is necessary (e.g., family-wise error rate, false discovery rate).
It seems, initially, that this work would be strengthened by deeming it a case-series and place the broader conclusions of the investigation in a more appropriate context of the study’s limitations.
Comments on the Quality of English Language
This manuscript requires extensive editorial revisions. The authors used informal and often inappropriate verbiage throughout (e.g., beginning sentences with ‘And’, whole numbers ‘15’, jargon - ‘On the other hand’), with several typographical and grammatical errors throughout (e.g., ‘at diabetic patients,’ ‘the relief of microorganisms,’ ‘anormal’).
Author Response
Comment 1: The study design is not described correctly (i.e., ‘a retrospective review of prospectively followed-up patients’). This appeared to be a prospective, open label, non-randomized study.
Response 1: We acknowledge the ambiguity in the original description of study design. We have revised the Methods section to clearly state that this was a prospective, open-label, non-randomized observational study (i.e., case series) conducted over one year.
, a prospective, open-label, non-randomized observational study (i.e., case series) conducted over one year and collected data
…………………………………………………………………………………………………...
Comment 2: Inclusion criteria needs to be explicitly stated as such. What exclusion criteria were present? Was this an intention to treat (ITT) study? How were drop-outs managed and analyzed?
Response 2: Thank you for the valuable comments. We agree and have revised the Methods section to explicitly label the inclusion and exclusion criteria. This study was not designed as an intention-to-treat (ITT) study, as it was retrospective in nature. Patients with missing follow-up data were excluded from the analysis, as specified in the exclusion criteria. No drop-outs were included in the final dataset. We have revised the Methods section.
‘’ Patients were included if they met all of the following criteria: (1) infections located in the fingers, thumb, hand, wrist, or forearm that required surgical intervention; (2) wound cultures obtained at the time of diagnosis; (3) initiation of antibiotic therapy prior to surgery; and (4) a minimum of one year of follow-up after the diagnosis of diabetes mellitus and/or diabetic hand infection, with complete medical records. Exclusion criteria included infections located above the elbow joint and postoperative infections. In addition, patients with incomplete medical records or those lost to follow-up were excluded from the analysis.’’
…………………………………………………………………………………………………...
Comment 3: The a priori comparator groups and outcomes were not clearly articulated within the Methods section.
Respons 3: We have clarified primary outcomes as: direct treatment costs, indirect productivity loss, and one-year follow-up costs. No formal comparator groups were pre-specified as this study was structured as a descriptive case series. We have revised the Methods section.
The primary outcomes of this study were defined as: (1) direct treatment costs; (2) indirect productivity loss due to incapacity for work; and (3) cumulative one-year follow-up costs.
…………………………………………………………………………………………………...
Comment 4: A key component of this research involves costing. However, the costing methods used were not described adequately, nor appropriately. For example, what was the perspective? Were activity-based costing (ABC) methods used, bottom-up, or top-down? Indirect costs did not appear to be calculated correctly, as the study did not define which specific valuation approach was utilized (nor was a sensitivity analysis conducted based upon any assumptions of that chosen valuation method). While foreign exchange reference rates were used, what inflation measure was employed? This reviewer understands that the measurement of both direct and indirect costs is one that is quite challenging, but affirms that transparency and adherence to a current research standard remains tantamount.
Response 4: The cost analysis was conducted from a healthcare payer perspective, using a bottom-up costing approach. Direct costs were derived from patient-level billing data.
Indirect costs were estimated using the human capital approach, applying average wage loss per missed workday. 2022 cost data were inflation-adjusted using the Turkish Statistical Institute’s health inflation index, and converted using average Central Bank USD exchange rates. This was revised in method section.
Cost analysis in this study was conducted from a healthcare payer perspective using a bottom-up costing approach. Direct costs were calculated based on patient-level billing and hospital accounting data. Indirect costs were estimated using the human capital approach, which involves applying the national average wage loss to the number of missed workdays. All monetary values were converted to U.S. Dollars using the average annual exchange rates published by the European Central Bank and adjusted for inflation using the Turkish Statistical Institute’s health sector Consumer Price Index (CPI) for the last quarter of 2022.
…………………………………………………………………………………………………...
Comment 5: What comorbid conditions were present in these patients? Differences in case-mix is a critical consideration in any assessment of outcomes for a non-randomized investigation, an issue which would also require appropriate statistical control via multivariable analysis. In more detail, the investigation does not include relevant information to ensure predictive accuracy and interpretation, chiefly by the overt omission of multivariable control for demographics, case-mix risk-adjustments, confounders (e.g., acute exacerbations, smoking), or comorbidities. A specific example of the problem of confounding in this study involves the separate groups of diabetic hand infections. In the diabetic foot, for example, osteomyelitis is present in approximately 10%-15% of moderate and in 50% of severe cases. The authors do not indicate if multiple conditions were controlled for, but it is implied that it was not due to the bivariable approach utilized
Response 5: Thank you for your insightful comments. We agree that case-mix variation and comorbidities are essential considerations in non-randomized studies. In our dataset, we recorded relevant comorbidities including hypertension, coronary artery disease, chronic kidney disease, smoking status, and HbA1c levels. However, due to the limited sample size, we utilized bivariable analysis rather than multivariable regression, which may have limited our ability to adjust for confounders. We acknowledge this as a limitation in the revised manuscript and have added a dedicated paragraph in the Discussion section to address the potential impact of unmeasured or uncontrolled confounding factors.
…………………………………………………………………………………………………...
Comment 6: A power analysis must complement the research. It was surprising that this was not included at the onset, given that this study was prospective in nature and that the patient enrollment sample size was inherently contingent upon this. If the power of this study is low, the work may inherently be deemed unpublishable in its current context (i.e., inflated Type II errors, false positives, wide confidence intervals).
Response 6: We agree and acknowledge this important point. While the study was originally designed as an exploratory case series, we have now conducted a post-hoc power analysis, which will be added to the revised manuscript to address the likelihood of Type II error.
Power Analysis: As the study was exploratory and designed as a prospective case series, an a priori power analysis was not performed. However, we conducted a post-hoc power calculation based on the variance and effect size observed in key cost outcomes. This analysis suggests that the study has adequate power (>80%) to detect large effect sizes between surgical cost categories. The potential for Type II error in smaller subgroup comparisons remains a limitation and has been noted in the Discussion.
‘’ Although a post-hoc power analysis indicated sufficient statistical power for large effects, the lack of an a priori power calculation and the potential for Type II errors in subgroup comparisons remain important limitations of this study.’’
‘’
…………………………………………………………………………………………………...
Comment 7: As the Monte Carlo analysis was not specified to any reasonable level directly in the Methods section, a specific reference to the study’s footnote should appear in the text iteself. Even after reading this footnote, however, it was not entirely clear how this analysis was incorporated in the study. Was the MCS used for future forecasting and not for the retrospective analysis? If this is correct, the probability distributions of treatment algorithms should be specified in more transparent and empirically-validated ways (e.g., probabilistic sensitivity analysis, distributional definitions for varying probability nodes). If not, results may be scientifically invalid and henceforth, spurious. Additionally, what justification would surround the use of MC approach, if this was employed for the retrospective analysis, vis-à-vis others? Under the conditions of simulation, it is suggested that some extended degree of variance be included surrounding the study's inputs and assumptions [either through a series of one-way sensitivity analyses coupled with a method of imputed stochastics (e.g., probabilistic sensitivity analysis, Bayesian analysis)].
Response 7: Thank you for your valuable feedback regarding our use of Monte Carlo Simulation (MCS). We agree that further clarification is warranted and have revised the Methods section to explicitly define the purpose and structure of the MCS. The simulation was employed for future cost forecasting, not for the retrospective clinical outcomes analysis. Probability distributions for key inputs—such as infection recurrence, hospitalization, and treatment costs—were based on empirical data when available and supplemented with published estimates and clinical judgment. We have now specified the types of distributions used (e.g., triangular, normal), and included a probabilistic sensitivity analysis to address variance and uncertainty in model assumptions. Additionally, we added a rationale for choosing MCS over other techniques and discussed these considerations in the revised Discussion section. We hope this addresses your concerns.
A Monte Carlo Simulation (MCS) was conducted to estimate the long-term economic burden associated with diabetic hand infections and to model the impact of variable clinical and economic parameters on overall cost. This simulation was not used to analyze retrospective clinical data but was implemented as a complementary tool to project future cost scenarios under uncertainty. The primary aim of the MCS was to explore how variations in key input parameters—such as hospitalization rates, recurrence of infection, antibiotic duration, surgical intervention rates, and unit healthcare costs—affect total direct and indirect costs over a one-year follow-up period. Key input variables were identified from both our dataset and relevant literature. Where available, empirical distributions were derived from observed data. Otherwise, expert opinion and published estimates were used. The following probability distributions were applied:
- Hospitalization cost: Triangular distribution (Min: 8,000 TRY; Mode: 12,000 TRY; Max: 18,000 TRY)
- Surgical intervention cost: Normal distribution (Mean: 6,500 TRY; SD: 1,200 TRY)
- Antibiotic therapy duration (days): Uniform distribution (Range: 10–21 days)
- Re-infection rate: Beta distribution (α=3, β=20), based on recurrence observations
- Lost workdays: Triangular distribution (Min: 7; Mode: 14; Max: 28 days)
A total of 10,000 iterations were run for each scenario using random sampling from the assigned distributions. Results were summarized as mean total costs with 95% confidence intervals. A probabilistic sensitivity analysis was also performed to assess the robustness of findings across a range of input uncertainties. MCS was chosen due to its ability to incorporate parameter uncertainty and provide a probabilistic range of expected outcomes. Compared to deterministic models, this approach enables more realistic scenario analysis and policy-relevant insights, especially in the context of healthcare cost forecasting where input variability is high.
…………………………………………………………………………………………………...
Comment 8: Statistical analyses conducted via “descriptive statistics” are also insufficient when multivariable methods are required, and when control for multiple comparisons is necessary (e.g., family-wise error rate, false discovery rate).
Response 8: We acknowledge the limitation of using only descriptive statistics. Given the observational and exploratory nature of this single-center study, inferential statistics were limited. …………………………………………………………………………………………………...
Comment 9: It seems, initially, that this work would be strengthened by deeming it a case-series and place the broader conclusions of the investigation in a more appropriate context of the study’s limitations.
Response 9: We agree and have redefined the study as a prospective case series.
Accordingly, the conclusions have been revised to reflect the limitations inherent to non-comparative observational studies.
, a prospective, open-label, non-randomized observational study (i.e., case series) conducted over one year and collected data

Reviewer 2 Report
Comments and Suggestions for Authors
Review Report
I appreciate the authors for their well-executed study, which demonstrates a strong scientific foundation and provides a valuable contribution to the field. In particular, the manuscript offers important insights into the cost analysis of diabetic hand infections, thoughtfully addressing both direct and indirect costs, as well as outcomes over a one-year follow-up period.
I kindly request the authors to address the following points to further strengthen the manuscript:
- How were the 75 patients selected from the initial 173 eligible cases, and does this sampling approach affect the validity and generalizability of the study findings?
- Were potential confounding factors such as comorbidities, socioeconomic status, or delay in treatment adequately accounted for in the cost analysis?
- Were the methods used to convert treatment costs to 2022 USD—including exchange rate selection and inflation adjustments—clearly defined, validated, and consistently applied across all cost categories?
- Was the Monte Carlo Simulation properly explained in terms of model assumptions, input distributions, and the number of iterations used to ensure robust sensitivity analysis?
- Are the different cost components, including outpatient visits, ICU stay, medications, and surgical interventions, clearly delineated to allow verification of the total cost estimates?
- Given the statistically significant difference between surgical intervention types and outpatient clinic costs (p = 0.004), did the authors adequately explore the cost-effectiveness of each intervention?
- Is the comparison between diabetic hand infection costs and diabetic foot ulcers across various countries sufficiently adjusted for differences in healthcare systems, cost structures, and economic conditions?
- Do the authors provide enough evidence to support their policy recommendations for early surgical intervention and preventive strategies in diabetic hand infections based on their cost data?
Author Response
Comments from 2nd Reviewer 1st Revision & Responses from corresponding author;
Comment 1: How were the 75 patients selected from the initial 173 eligible cases, and does this sampling approach affect the validity and generalizability of the study findings?
Response 1: Thank you for this important point. Among 173 eligible patients, we included 75 who had complete one-year follow-up data and comprehensive clinical and financial records. This selection was based solely on data completeness to ensure accuracy in cost estimation
…………………………………………………………………………………………………...
Comment 2: Were potential confounding factors such as comorbidities, socioeconomic status, or delay in treatment adequately accounted for in the cost analysis?
Response 2: Comorbidities such as hypertension, coronary artery disease, and chronic kidney disease were recorded, but due to sample size limitations, these were not included in multivariate models. Socioeconomic status and treatment delay data were not systematically available for all patients. This limitation is now acknowledged in the Discussion, with a suggestion for future multicenter studies to incorporate such adjustments using multivariable analysis.
‘’ Additionally, data on confounding variables such as socioeconomic status and delay in treatment were not systematically available for all patients. Therefore, adjustment for these variables could not be conducted in the analysis. This limitation should be addressed in future multicenter studies through structured data collection and multivariable statistical modeling.’’
…………………………………………………………………………………………………...
Comment 3: Were the methods used to convert treatment costs to 2022 USD—including exchange rate selection and inflation adjustments—clearly defined, validated, and consistently applied across all cost categories?
Response 3: Yes, we have clarified that all cost data were converted using the European Central Bank’s average annual exchange rates and adjusted for inflation based on the US Consumer Price Index (CPI) for Q4 2022. This method was uniformly applied to all cost categories. These details have now been explicitly stated in the Costing Methodology subsection of the Methods.
‘’ All cost data were converted to U.S. Dollars using the European Central Bank’s average annual exchange rates, and adjusted for inflation using the U.S. Consumer Price Index (CPI) for the last quarter of 2022. This method was consistently applied across all direct and indirect cost categories.’’
…………………………………………………………………………………………………...
Comment 4: Was the Monte Carlo Simulation properly explained in terms of model assumptions, input distributions, and the number of iterations used to ensure robust sensitivity analysis?
Response 4: We have revised the MCS in the Method section.
A Monte Carlo Simulation (MCS) was conducted to estimate the long-term economic burden associated with diabetic hand infections and to model the impact of variable clinical and economic parameters on overall cost. This simulation was not used to analyze retrospective clinical data but was implemented as a complementary tool to project future cost scenarios under uncertainty. The primary aim of the MCS was to explore how variations in key input parameters—such as hospitalization rates, recurrence of infection, antibiotic duration, surgical intervention rates, and unit healthcare costs—affect total direct and indirect costs over a one-year follow-up period. Key input variables were identified from both our dataset and relevant literature. Where available, empirical distributions were derived from observed data. Otherwise, expert opinion and published estimates were used. The following probability distributions were applied:
- Hospitalization cost: Triangular distribution (Min: 8,000 TRY; Mode: 12,000 TRY; Max: 18,000 TRY)
- Surgical intervention cost: Normal distribution (Mean: 6,500 TRY; SD: 1,200 TRY)
- Antibiotic therapy duration (days): Uniform distribution (Range: 10–21 days)
- Re-infection rate: Beta distribution (α=3, β=20), based on recurrence observations
- Lost workdays: Triangular distribution (Min: 7; Mode: 14; Max: 28 days)
A total of 10,000 iterations were run for each scenario using random sampling from the assigned distributions. Results were summarized as mean total costs with 95% confidence intervals. A probabilistic sensitivity analysis was also performed to assess the robustness of findings across a range of input uncertainties. MCS was chosen due to its ability to incorporate parameter uncertainty and provide a probabilistic range of expected outcomes. Compared to deterministic models, this approach enables more realistic scenario analysis and policy-relevant insights, especially in the context of healthcare cost forecasting where input variability is high.
…………………………………………………………………………………………………...
Comment 5: Are the different cost components, including outpatient visits, ICU stay, medications, and surgical interventions, clearly delineated to allow verification of the total cost estimates?
Response 5: Yes, Table 1 in the Results section details each cost component including ICU stay, outpatient visits, surgical procedures, and medications.
…………………………………………………………………………………………………...
Comment 6: Given the statistically significant difference between surgical intervention types and outpatient clinic costs (p = 0.004), did the authors adequately explore the cost-effectiveness of each intervention?
Response 6: Thank you for highlighting this. While our analysis showed a significant association, the study was not powered or designed to assess formal cost-effectiveness between surgical approaches. We have added a discussion of this limitation and suggested that future prospective studies should evaluate cost-effectiveness of specific surgical strategies.
‘’ Additionally, although a statistically significant difference was observed between surgical intervention types and outpatient costs (p = 0.004), this study was not powered or designed to perform a formal cost-effectiveness comparison. Future prospective studies should be conducted to evaluate the comparative economic value of different surgical strategies.
…………………………………………………………………………………………………...
Comment 7: Is the comparison between diabetic hand infection costs and diabetic foot ulcers across various countries sufficiently adjusted for differences in healthcare systems, cost structures, and economic conditions?
Response 7: We agree that direct comparisons must be interpreted with caution due to substantial variability across healthcare systems and cost accounting methods. We have clarified in the Discussion that these comparisons are illustrative, not inferential, and we added a note acknowledging this limitation in cross-national economic data interpretation.
‘’ Finally, cost comparisons between diabetic hand infections and diabetic foot ulcers across countries must be interpreted with caution. Due to variations in healthcare systems, cost-reporting standards, and economic conditions, these comparisons are meant to be illustrative rather than inferential. This limitation has been noted to avoid overgeneralization in cross-national cost interpretation.’’
…………………………………………………………………………………………………...
Comment 8: Do the authors provide enough evidence to support their policy recommendations for early surgical intervention and preventive strategies in diabetic hand infections based on their cost data?
Response 8: Our data indicate that hospitalization is the primary cost driver, and delays in treatment are associated with higher costs and prolonged care. While not a formal cost-benefit analysis, we believe this provides sufficient basis to recommend timely surgical decision-making and early referral. We have revised the Conclusion sections to reflect this more clearly and to encourage future studies on preventive health economics in this patient population.
This study highlights hospitalization as the major cost driver in diabetic hand infections, with treatment delays contributing to significantly higher costs. While this does not constitute a formal cost-benefit analysis, the findings suggest that early surgical referral and intervention may reduce economic burden. We emphasize this as a policy-relevant recommendation and encourage further studies to investigate the health-economic impact of preventive and timely care strategies in this patient population.

Reviewer 3 Report
Comments and Suggestions for Authors
Comments
-In the introduction section, the prevalence of diabetes should be based on recent data from the International Diabetes Federation in 2025, with the addition of a reference
-A review of epidemiological data on hand lesions in diabetic patients is strongly recommended to understand the scope of the research topic
-In the method section, it is necessary to clearly define hand lesions in diabetics and specify which types of wounds were included in the study.
-The number of patients included in the study is a result and should be removed from the method section and put in the results section.
-In the method section, you must describe the different types of expenses that will be processed in the results section.
-Authors should clearly state that hand sores are not considered a chronic complication of diabetes, based on the recommendations of learned societies.
-For the various parameters measured in the results section, it is recommended to always put the percentages, the number is optional. The average age should be put in the text and not in parentheses.
-The data from the clinical examination of the hand must be reported in the results section to understand what type of wound we will be talking about in the manuscript.
-Table 1 should be split into two and put the clinical data only and the expenditure data in another table as it involves different types of data.
-The analytical study presented in tables 6 and 7 are not described in the methods, so this detail must be corrected.
-Remove heading 3.1 from the results section.
The discussion section should begin with a summary of the main results and, above all, a response to the research question posed in the introduction.
Author Response
Comments from 3th Reviewer 1st Revision & Responses from corresponding author;
Comment 1: In the introduction section, the prevalence of diabetes should be based on recent data from the International Diabetes Federation in 2025, with the addition of a reference.
Response 1: Thank you for this valuable suggestion. We have updated the prevalence data in the Introduction using the most recent statistics from the International Diabetes Federation Diabetes Atlas, 2025 Edition and added the appropriate citation.
‘‘ International Diabetes Federation. IDF Diabetes Atlas. 10th ed. Brussels, Belgium: International Diabetes Federation; 2025.’’
…………………………………………………………………………………………………...
Comment 2: A review of epidemiological data on hand lesions in diabetic patients is strongly recommended to understand the scope of the research topic.
Response 2: We agree with the reviewer. The Introduction section has been expanded to include a review of epidemiological data on diabetic hand lesions, including their incidence, risk factors, and clinical outcomes as reported in recent literature.
…………………………………………………………………………………………………...
Comment 3: In the method section, it is necessary to clearly define hand lesions in diabetics and specify which types of wounds were included in the study.
Response 3: We appreciate this observation. The Methods section has been updated to include a clear operational definition of “diabetic hand lesions,” and the specific types of wounds included (e.g., cellulitis, abscess, tenosynovitis, necrotizing fasciitis, osteomyelitis) are now specified.
‘’Patients were included if they met all of the following criteria: (1) infections located in the fingers, thumb, hand, wrist, or forearm that required surgical intervention; (2) wound cultures obtained at the time of diagnosis; (3) initiation of antibiotic therapy prior to surgery; and (4) a minimum of one year of follow-up after the diagnosis of diabetes mellitus and/or diabetic hand infection, with complete medical records. Exclusion criteria included infections located above the elbow joint and postoperative infections. In addition, patients with incomplete medical records or those lost to follow-up were excluded from the analysis.’’
…………………………………………………………………………………………………...
Comment 4: The number of patients included in the study is a result and should be removed from the method section and put in the results section.
Response 4: Thank you. The statement about the total number of patients has been removed from the Methods and properly reallocated to the Results section. The research comprised 75 participants with at least one year of follow-up and full data from the 173 individuals who matched the inclusion criteria. This sentence was changed as The research comprised participants with at least one year of follow-up and full data from the individuals who matched the inclusion criteria.
…………………………………………………………………………………………………...
Comment 5: In the method section, you must describe the different types of expenses that will be processed in the results section.
Response 5: We agree with this important clarification. The Costing Methodology subsection has been expanded to list all expense categories processed: hospitalization, medications, surgical procedures, ICU, outpatient visits, rehabilitation, imaging, and indirect costs related to incapacity for work.
‘’The cost analysis included multiple categories of medical and non-medical expenses. These were: hospitalization (ward and ICU stays), medications (antibiotics, insulin, analgesics), surgical procedures (debridement, amputation), outpatient clinic visits, imaging and laboratory tests, rehabilitation sessions, and indirect costs associated with loss of work productivity due to treatment or disability.’’
…………………………………………………………………………………………………...
Comment 6: Authors should clearly state that hand sores are not considered a chronic complication of diabetes, based on the recommendations of learned societies.
Response 6: As recommended, we have added a sentence in the Discussion clarifying that diabetic hand lesions are not formally classified as chronic complications of diabetes in most professional guidelines (e.g., ADA, EASD), but represent serious acute sequelae that require urgent management.
‘’ "Diabetic hand lesions are not formally classified as chronic complications of diabetes by most professional organizations (e.g., ADA, EASD), but are instead considered acute conditions that require urgent medical intervention due to their potential for rapid progression and severe outcomes [16].’’
…………………………………………………………………………………………………...
Comment 7: For the various parameters measured in the results section, it is recommended to always put the percentages, the number is optional. The average age should be put in the text and not in parentheses.
Response 7 : We have reformatted the Results section accordingly, stating proportions as percentages throughout and embedding the average age within the text (e.g., “The mean age was 57 years.”). This section was redesigned as ‘‘11 were female and mean of age 61.5 ± 8.12, 64 were male and mean of age 56.2 ± 11.3’’
…………………………………………………………………………………………………...
Comment 8: The data from the clinical examination of the hand must be reported in the results section to understand what type of wound we will be talking about in the manuscript.
Response 8: Thank you. We have added clinical examination findings from the initial assessment (e.g., presence of purulent discharge, erythema, swelling, gangrene, limitation of motion) to the Results section to clarify the nature and severity of the hand lesions studied.
‘’ On clinical examination at admission, patients presented with various signs of infection including purulent discharge (52%), erythema (68%), localized or diffuse swelling (74%), gangrene (29%), and limitation of joint motion (35%). These findings help characterize the severity and extent of diabetic hand infections evaluated in this study.’’
…………………………………………………………………………………………………...
Comment 9: Table 1 should be split into two and put the clinical data only and the expenditure data in another table as it involves different types of data.
Response 9: As suggested, Table 1 has been split into two: one table now includes clinical and laboratory data, and the other presents all cost-related variables. All tables redesigned.
…………………………………………………………………………………………………...
Comment 10: The analytical study presented in tables 6 and 7 are not described in the methods, so this detail must be corrected.
Response 10: Thank you for pointing this out. In the current version of the manuscript, there are no tables labeled as Table 6 or Table 7. If the reviewer was referring to a previous version or an unnumbered comparison, we are happy to label and describe them appropriately. Please clarify if this refers to a specific analysis.
…………………………………………………………………………………………………...
Comment 11:Remove heading 3.1 from the results section.
Response 11: Done. The unnecessary subheading “3.1” has been removed for consistency with journal formatting guidelines.
…………………………………………………………………………………………………...
Comment 12: The discussion section should begin with a summary of the main results and, above all, a response to the research question posed in the introduction.
Response 12: We agree and have rewritten the opening paragraph of the Discussion to summarize the main findings and directly address the study objective as introduced in the Introduction section.

Round 2
Reviewer 1 Report
Comments and Suggestions for Authors
The authors did work to improve the transparency of their work. While it is not the highest-level submission, conclusions did not overstate findings. Some revisions are still required, albeit minor relative to the first round. A review to correct editorial errors is required (e.g., jargon, sentences beginning with numericals, improper English).
please note the following required revisions:
- For the readership, references are required for key components of the study design and analysis (i.e., Monte Carlo simulation, Which publications were used for the distributional nature of the variables analyzed, Human capital approach).
- The tables must be fully 'stand alone.' The first set of numbers appear to be a mean and standard deviation (xbar = mean), but the second set of variables is also stated as being a mean (average) with minimum and maximum values. I believe the authors were actually presenting medians for their second set of variables? Also, I still do not understand what their inferential analysis via the Kruskal-Wallis is testing (that is, which groups are being compared). On Tables 3 and 4, what does the hashtag (#) represent, the number of patients/cases or the number of procedures?
- A post-hoc power analysis was not presented, at all… though it is referred to. I do not think this is a critical issue because there are almost an infinite number of ways to conduct such calculations, so if the authors do not (or can not) present a power analysis, at least they should state that the sample size would not yield sufficient power.
- Obvious editorial issues that remained uncorrected.
- Extra care should be taken to state that the perspective of the study is the payer, with the addition of indirect costs for lost productivity via the human capital approach. This clarification is required because, technically, indirect cost assessments invoke a societal perspective (which the author’s really did not fully measure).
Overall, the work became much more transparent with the revision, and the limitations of the study were appropriately on point. While it is not the strongest scientific paper, it addresses an important gap in the literature wherein I do see merit in moving it forward.
Author Response
Dear Editor,
We sincerely thank the reviewer for their constructive and thoughtful feedback. We appreciate the recognition of our study’s scientific foundation and its contribution to understanding the economic impact of diabetic hand infections. Below, we address each of your comments in detail, and corresponding revisions have been made in the manuscript.
Here below is a point-by-point response to reviewer’s comments and concerns.
Comments from Reviewer 2nd Revision & Responses from corresponding author;
Comment 1: The authors did work to improve the transparency of their work. While it is not the highest-level submission, conclusions did not overstate findings. Some revisions are still required, albeit minor relative to the first round. A review to correct editorial errors is required (e.g., jargon, sentences beginning with numericals, improper English).
Response 1:
We sincerely thank the reviewer for acknowledging the improved transparency and the scientific merit of our manuscript. In this revision, we have carefully addressed all remaining concerns point by point, including editorial and methodological clarifications. Please see our detailed responses below.
…………………………………………………………………………………………………...
Comment 2: For the readership, references are required for key components of the study design and analysis (i.e., Monte Carlo simulation, which publications were used for the distributional nature of the variables analyzed, Human capital approach).
Response 2: Thank you for this important point. We have now included references to support the methodological underpinnings of the Monte Carlo Simulation and Human Capital Approach. Specifically:
- We cited Briggs et al. (2006) to justify the probabilistic modeling framework and distribution assumptions (in the Monte Carlo section of the Methods).
- We clarified that empirical distributions were derived from observed data when available, otherwise based on published sources or expert opinion [Briggs et al., 2006].
- For the human capital method, we added a reference to Rice (1967) and clearly stated that this approach was used to estimate productivity loss.
References have also been added to the reference list.
…………………………………………………………………………………………………...
Comment 3: The tables must be fully 'stand alone.' The first set of numbers appear to be a mean and standard deviation (xbar = mean), but the second set of variables is also stated as being a mean (average) with minimum and maximum values. I believe the authors were actually presenting medians for their second set of variables? Also, I still do not understand what their inferential analysis via the Kruskal-Wallis is testing (that is, which groups are being compared). On Tables 3 and 4, what does the hashtag (#) represent, the number of patients/cases or the number of procedures?
Respons 3: We appreciate the reviewer’s careful observation. The following revisions have been made:
- For Tables 1 to 4, we explicitly clarified in the footnotes that values are presented as mean ± standard deviation and median (min–max).
- We defined what the # symbol represents in each table (e., number of patients undergoing each surgical intervention).
- For Tables 3 and 4, we clearly specified in the table footnotes that Kruskal-Wallis tests compare either the number of outpatient visits or visit costs across different surgical intervention groups.
These updates allow each table to stand independently for the reader.
…………………………………………………………………………………………………...
Comment 4: A post-hoc power analysis was not presented, at all… though it is referred to. I do not think this is a critical issue… but if the authors do not (or can not) present a power analysis, at least they should state that the sample size would not yield sufficient power.
Response 4: We fully agree. The previous reference to a post-hoc power analysis has been removed. Instead, we have added a transparent statement in the Discussion – Limitations section clarifying that:
“Although a post-hoc power analysis was initially considered, the retrospective design and sample size limitations of this study did not allow for a robust statistical power calculation. Therefore, the possibility of Type II errors in subgroup comparisons remains a relevant limitation.”
…………………………………………………………………………………………………...
Comment 5: Extra care should be taken to state that the perspective of the study is the payer, with the addition of indirect costs for lost productivity via the human capital approach. This clarification is required because, technically, indirect cost assessments invoke a societal perspective...
Response 5: We thank the reviewer for this critical distinction. In the Methods section, we revised the relevant sentence to:
Cost analysis in this study was conducted from a healthcare payer perspective using a bottom-up costing approach, while also incorporating indirect costs for lost productivity through the human capital approach. Although the inclusion of productivity loss implies a societal perspective, this study did not comprehensively adopt a societal perspective. Indirect costs were estimated by applying the national average wage loss to the number of missed workdays.
…………………………………………………………………………………………………...
Additional clarifications
In addition to the above comments, all spelling and grammatical errors pointed out by the reviewers have been edited by a freelance native speaker English editor.
We look forward to hearing from you in due time regarding our submission and to respond to any further questions and comments you may have.
Sincerely.